# The selection force weakens with age because ageing evolves and not vice versa

Stefano Giaimo [1✉] & Arne Traulsen [1]

According to the classic theory of life history evolution, ageing evolves because selection on traits necessarily weakens throughout reproductive life. But this inexorable decline of the selection force with adult age was shown to crucially depend on specific assumptions that are not necessarily fulfilled. Whether ageing still evolves upon their relaxation remains an open problem. Here, we propose a fully dynamical model of life history evolution that does not presuppose any specific pattern the force of selection should follow. The model shows: (i) ageing can stably evolve, but negative ageing cannot; (ii) when ageing is a stable equilibrium, the associated selection force decreases with reproductive age; (iii) non-decreasing selection is either a transient or an unstable phenomenon. Thus, we generalize the classic theory of the evolution of ageing while overturning its logic: the decline of selection with age evolves dynamically, and is not an implicit consequence of certain assumptions.

[1] Department of Evolutionary Theory, Max Planck Institute for Evolutionary Biology, August-Thienemann-Straße 2, 24306 Plön, Germany.
✉email: giaimo@evolbio.mpg.de

Ageing is a degeneration in the physiological state of adult individuals that progressively curbs their fecundity and survival as their age increases[1,2]. Ageing is observed in a great number of species[3–5]. Why does ageing evolve? Established theories[1,6–9] state that this is because the force of selection suffers an inexorable decline with adult age. Accordingly, the fitness value of reproduction and survival late in life is discounted by a combination of two factors. One is the inherent improbability, from the point of view of a newborn, of reaching late ages when compared to any earlier age. The other factor is a dilution effect on late reproduction, should the population be growing in size. Therefore, the likelihood of detrimental mutations, with or without earlier pleiotropic benefits, to spread and persist in the population would increase the later their age of action. Ageing would be the phenotypic manifestation of such mutations.

The pillar of this established view was undermined by what has been deemed as one of the major insights in the field over the last 35 years: the force of selection may not necessarily decline with adult age[10]. In particular, Baudisch put Hamilton's model[8], recognized as the most mathematically explicit version of the classic theory[1,11,12], under scrutiny. This revealed that a necessary decline in the selection force with age only derives from the restrictive assumptions that, when mutations act at each age separately, they do so via small additive changes in fecundity and small proportional changes in survival[10,13,14]. Under alternative and equally plausible assumptions about the working of age-specific mutations, the selection force was shown to increase with age in some cases[13]. For example, reversing the assumptions of the classic theory, small proportional changes in fecundity and small additive changes in survival can lead to increasing selection over some ages (Fig. 1). In other words, Baudisch's work showed that the classic theory implicitly assumed an inexorably declining selection force with age rather than deriving it from general principles. To what extent this revelation undermines the validity of the classic theory remains unclear. This is the starting point of the present work. Our goal is an in-depth exploration of the consequences of possibly non-declining selection with age on life history evolution.

The strength of selection on fecundity and survival at a given age depends on the whole life history[15,16], i.e., on the complete age trajectories of fecundity and survival. The age-specific selection force then typically changes as the life history evolves. The classic theory, however, identifies a pattern to this force that is invariant to evolutionary change. Under specific assumptions about how mutations alter fecundity and survival, there is a persistent decline in a selection over adult ages that is independent of the schedules of fecundity and survival[8]. Ecological factors, like predators and environmental hazards, that impact population density can modify selective forces that mold the life history[6,17–19]. But these factors can at most modulate, and not counter, the decline of selection with adult age. The exact quantification of age-specific fecundity and survival, as well as of the corresponding selective forces and their change during evolution, is then irrelevant to the conclusion of the classic theory that ageing is evolutionary inevitable. The invariant bias of selection against late-life alone implies this conclusion.

However, once we relax the assumptions of the classic theory, selection may not decline with reproductive age. And there no

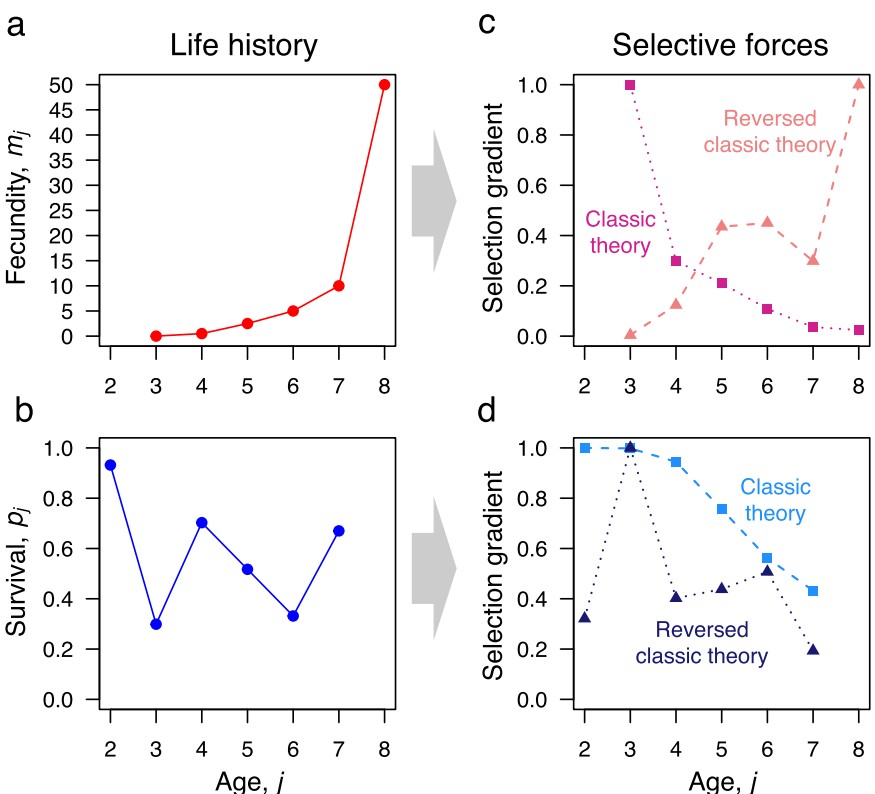

**Fig. 1 Selection gradients on fecundity and survival under additive and proportional effects. a, b** Fecundity $m_j$ at age $j$ and survival $p_j$ from age $j$ to $j+1$ are reported for a hypothetical life history with first reproduction at age 3 and maximum age 8. Survival at prereproductive ages is $p_0 = 0.408$ and $p_1 = 0.943$. **c, d** Selection gradients for this life history are computed on both fecundity and survival under both additive (dotted line) and proportional (dashed line) genetic effects. The classic theory[8] of the evolution of ageing assumes additive effects on fecundity and proportional effects on survival, which lead to selection gradients that decline with reproductive age. These are here compared with the reversed case[13]: proportional effects on fecundity and additive effects on survival, which may lead to selection gradients that do not decline with reproductive age. Selection gradients are scaled relative to their maximum value.

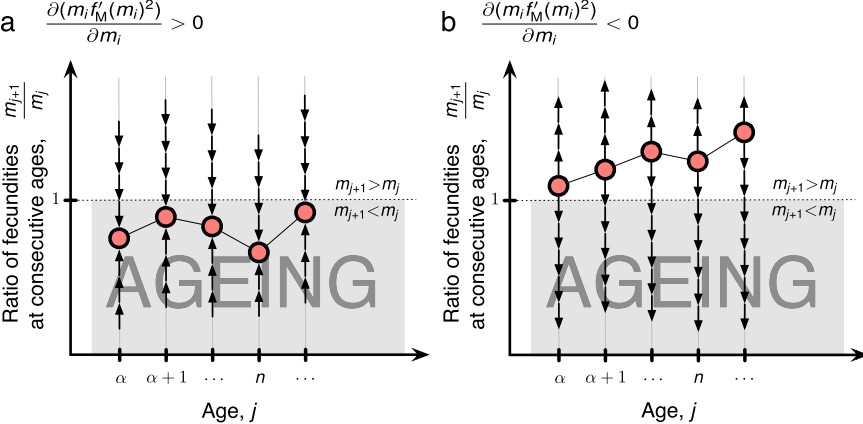

**Fig. 2 Evolutionary dynamics of fecundity. a, b** Dynamics of fecundity evolution depend on how genetic variation acts on fecundity $m_i$ at each age $i$. There are two main cases that are captured by the function $f_M$ (see Methods). **a** Dynamics, depicted here as arrows, always lead fecundity at one age to be larger than fecundity at the next age and there is at most a single equilibrium with this pattern, which is stable under broad conditions. **b** There may be at most a single equilibrium where fecundity at one age is smaller than fecundity at the next age, but dynamics always repel this equilibrium.

longer is an invariant pattern to the selection force that we can leverage. Any inference about the evolutionary inevitability of ageing is then unwarranted. The following questions naturally emerge: Can selection that increases with age persist with time? Can this lead to the evolution of increasing fecundity and survival with age (negative ageing)? Do the initial conditions, i.e., the ancestral life history, matter to the outcome of the evolutionary process? To answer these questions, we should keep track of the evolutionary dynamics of fecundity and survival at each age, i.e., how they change with time in response to current selection on them, and predict the long-run tendencies of these dynamics. Here, we propose and analyse a dynamic model that does just that. At the heart of our model is the breeder's equation from quantitative genetics[20]. We simplify this equation by assuming small additive genetic variance equal for all traits and no covariances between traits at all times. The model can accommodate all sorts of genetic effects beyond additive and proportional effects. But we continue to assume effects limited to single ages. In this way, the selection force is not constrained to decline with age. The model also includes a mechanism of density dependence to keep population size constant. The evolution of fecundity and the evolution of survival are studied separately.

## Results

Our model enables us to give a dynamical meaning to the notion of 'inevitability' that is often evoked with reference to the evolution of ageing[8]. Ageing, a progressive decline in fecundity and survival with reproductive age, is the only stable outcome of life history evolution under age-specific genetic variation, i.e. with effects at single ages. This is so even if such variation modifies fecundity and survival in ways that may lead to increasing selection over reproductive ages and largely irrespective of the ancestral life history from which the evolutionary process is initiated. An ageless life history, where fecundity and survival remain constant with age throughout the reproductive period, cannot be an equilibrium. Unexpectedly, negative ageing may be an equilibrium, but it is unstable. More generally, the absence of ageing is a transient or unstable state. The selection forces on fecundity and survival at reproductive ages may not always show a pattern of persistent decline during evolution. But, again, this is a transient or unstable phenomenon. When evolutionary dynamics reach a stable resting state and, therefore, ageing has evolved, equilibrium selective forces must display a pattern of

persistent decline with age. Results that are specific to fecundity evolution or to survival evolution are reported hereafter.

**Fecundity**. Under broad conditions, if an equilibrium exists, it is unique. Depending on the exact form of genetic effects, this equilibrium can take two forms. Starting from the age of first reproduction, either fecundity decreases with each successive age or fecundity increases with each successive age. In the former case (ageing), the equilibrium is attractive and stable, while in the latter case (negative ageing) the equilibrium is unstable and everywhere repelling (Fig. 2).

**Survival**. Life histories for which survival at each reproductive age is greater than survival at any later age (ageing) form a globally attractive set for evolutionary dynamics. Life histories for which survival increases with each reproductive age (negative ageing) instead form a set that these dynamics repel. Depending on the form of genetic effects, equilibrium survival may be found inside one or the other set. The existence and number of equilibria, as well as exact dynamics near them, remain unknown (Fig. 3). An analysis of a simplified model molded after[21] with only two age classes (Supplementary Methods) reveals that there may be more than one equilibrium at which survival displays a decline with age. Proportional effects on survival, as assumed by the classic theory, do not lead to an equilibrium in our model.

## Discussion

In our model, ageing is evolutionarily inevitable in a dynamical sense irrespective of the genetics of fecundity and survival. Selective forces may at times be stronger in late life than in earlier life. But, as we show, this property pertains to either a transient or an unstable state, which is eventually abandoned. An ever-declining force with age is not an intrinsic property of selection and the one driver behind the evolution of ageing, as the classic theory[8] implicitly assumes[13]. Instead, a persistent, age-related weakening of selective forces is itself a result of evolution. Our model may then be viewed as a generalization of the classic theory where an implicit assumption of the latter is turned into a prediction.

In this generalization process, some assumptions of the classic theory (type of genetic effects) are relaxed, while others are kept. An aspect of relevance is that our model, like the classic theory, presupposes genetic variation with effects that are limited to single ages. From the theoretical standpoint, the assumption of

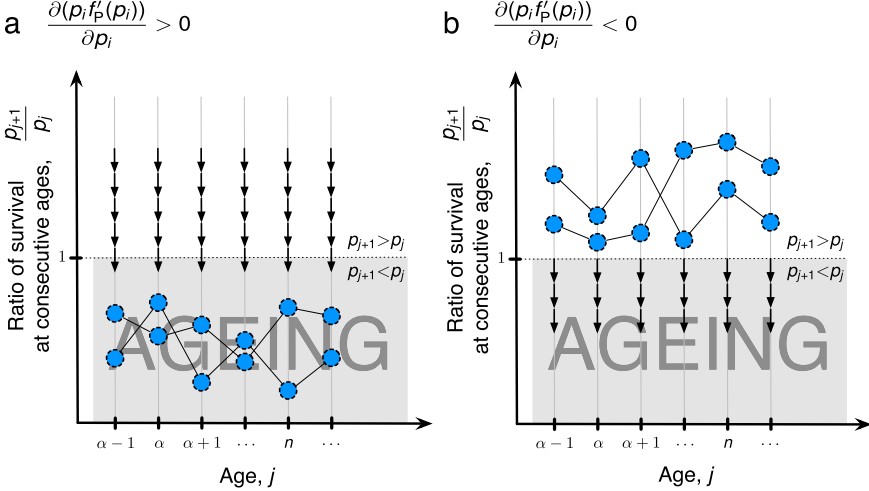

**Fig. 3 Evolutionary dynamics of survival. a, b** Dynamics of survival evolution depend on how genetic variation acts on survival $p_i$ from each age $i$ to the next age $i+1$. There are two main cases that are captured by the function $f_P$ (see Methods). Only survival that leads into reproductive ages is here considered with $\alpha$ the first reproductive age. The existence, number, and stability of equilibria remain undetermined in the general case. **a** Dynamics, depicted here as arrows, always lead survival at one age to be larger than survival at the next age. Any equilibrium survival schedule shows a decline of survival with each successive age. **b** If there are equilibrium schedules where survival increases with age, these equilibria are located within a region that dynamics always repel.

age-specific effects has found both critics[9,22–24] and supporters[25,26]. The experimental literature appears to be equally mixed[27]. While one study did not find evidence of age-specific effects[28], another study found evidence that these effects exist, yet not at all ages[29]. More recently, age-specific mutational effects with influence on fecundity were detected[30,31]. Research on model organisms shows how a single mutation may modify the organism's entire survival profile[32]. This fact alone, however, does not preclude the existence of age-specific mutations with an effect on survival. Overall, more experiments are needed to understand the age scope of mutations[33,34]. Importantly, effects on survival lasting indefinitely after the age of onset avoid the possibility that selection may increase with age[35].

However, age-specificity remains an important component of the most advanced developments of the mutation-accumulation hypothesis of ageing[25,26]. The newest models of this hypothesis have overcome previous linearity assumptions in dealing with epistatic interactions among mutations[36]. These models are sufficiently powerful to allow computation of, among other things, the equilibrium distribution, when it exists, of mortality over age in a genetically and, therefore, demographically heterogeneous population under mutation-selection balance[25,26]. Some equilibrium results of these models can explain features of observed mortality, i.e., its deceleration at very late ages[37], that are not within the reach of the classic theory. They also yield predictions, i.e., extremal mortality before the last reproductive age, that even contradict this theory. Despite sharing both an assumption of age-specificity and a dynamical perspective, a direct comparison of our work with these models appears arduous because of other, diverging assumptions in the overall approach. The architecture of advanced models of mutation-accumulation is very general[38]. They are genetically detailed in the haploid case. They can in principle accommodate both beneficial mutations and detrimental ones as well as effects that may be pleiotropic to some degree, i.e., of different sign at different ages, or not only additive. But analysis of these models appears to be mostly focused on the case of detrimental, age-specific mutations with additive effects on mortality. The aim is to compute the load of such mutations that a population can tolerate. Our model, instead, is not genetically explicit and tracks life history evolution at the phenotypic level. Selection only acts on beneficial variation with a typically non-additive effect on fecundity and mortality. Moreover, our analysis in this work is mostly qualitative. The focus is on the ultimate age pattern of fecundity and survival and their selection gradients, i.e., whether they increase, decline, or stay constant with age. We are not interested in the exact shape of these trajectories at equilibrium, although some of these may be computed with our methods (Supplementary Methods).

Our broadly qualitative look at age patterns and our presupposition of a perennial tendency to fitness increase are also behind our assumption, only alluded to in the classic theory[7,8], of density dependence. On the one hand, any model with a long-term view like ours should include a mechanism to keep population size in check. Otherwise, the fixation of beneficial variants would eventually make population growth unsustainable. On the other hand, the still ongoing debate on the role of density dependence in the presence of extrinsic mortality[6,17–19,35] shows that the exact form of density dependence a population experiences may matter to the evolution of ageing. Of those forms the debate has considered (our age-independent decrements in fecundity or survival being one of them), some may modulate the rate of ageing, i.e., how fast fecundity and survival decline with reproductive age. Therefore, we cannot exclude that the shape of equilibrium fecundity and survival schedules for our model may bear the signature of the assumed form of density dependence. But this issue is out of the scope of the present work. Here, we are concerned with the extent to which the conclusions of the classic theory hold in the face of genetic effects that can lead to increasing selection with age. However, when these effects are proportional, i.e., the classical assumption, they act on survival in the same way as our chosen mechanism of density dependence. This may be the reason that the classic theory has no equilibrium in our model.

A potential limitation of our results comes from our usage of the breeder's equation. With a constant variance-covariance structure among traits, the accuracy of this equation best approximates short-term evolutionary change only[39]. This is because the whole trait distributions are not tracked as they change along with the variability upon which selection acts. In this respect, we should remark again that our dynamical analysis is qualitative in nature, as we are interested in whether life

histories have a tendency to evolve ageing or not. We do not expect accurate, long-term predictions of equilibrium fecundity and survival at each age from our recursion of the breeder's equation. Obtaining the overall directions of life history evolution suffice. Moreover, our local stability analysis of equilibria should not be affected by the long-term inaccuracies that are inherent to the breeder's equation.

The evolution of the variance-covariance structure among traits, i.e., the $G$ matrix for a finite number of traits, is a major research topic that has proven hard to deal with analytically[40]. For our purposes, assuming such structure to be constant with no covariances and small variances for each trait is needed for analytical tractability. This is also an assumption of the classic theory. Including a non-trivial, time-dependent covariance structure in our model may thus modify our findings—but such an approach would likely come with substantial analytical challenges.

However, our results are still indirectly suggestive of what the role may be of the trade-offs that determine those trait covariances—and which we have excluded from our model. Looking across the tree of life, trajectories of fecundity and survival over adulthood are very diverse from one species to another[5]. Ageing is only one of the many patterns that these trajectories can follow. It has been argued[23] that the inability of the classic theory[8] to explain these observations could be due to two of its assumptions: (i) age-specific genetic effects that necessarily lead to declining selection with age and (ii) absence of trade-offs between fecundity and survival across different ages. Reasoning within this scheme, our work shows that, once trade-offs are kept out of the picture and any sort of age-specific genetic effect is allowed, ageing remains the one and only stable outcome of evolution. We are then left with the dilemma of whether the observed lack of ageing in certain branches of the tree of life should be regarded as a merely transient phenomenon, as our stylized model would suggest, or whether trade-offs can stabilize the lack of ageing. We favour this second option for two main reasons. First, dependencies among fitness components have been shown pervasive in life history evolution[41–43]. Second, the application of optimization principles has already demonstrated how these can account for a vast array of different life histories[14]. In theoretical models, the specific functional form of trade-offs was already shown to determine whether ageing or something other than ageing is an evolutionarily stable strategy[44–48]. However, optimization models are not dynamic in that they do not describe how an ancestral population evolves over time. It would be interesting to apply a dynamical perspective to models that structure the life history into stages (e.g., size classes, phases of development)[16]. In stage-structured models, trade-offs between ages are deeply ingrained[49], as a stage may lump together fecundity or survival over separate ages and could be under genetic control. These models, once equipped with dynamics, may give us a new perspective on the evolution of the trajectories of fecundity and survival over age[23].

The classic theory was deemed "heuristic"[50], presumably because of its lack of a dynamical dimension. Adding this dimension shows that attributing the evolution of ageing to a persistently declining force of selection with age is indeed a heuristic principle. But while insufficient to explain the various ways in which organisms are observed to allocate fecundity and survival in their lifespan, this principle is still valuable in giving us a glimpse into how ageing evolves. Violating the principle in fact does not affect the conclusion of the classic theory that when fecundity and survival evolve independently at each age, ageing is evolutionarily inevitable. However, the reasons behind this inevitability appear more subtle than originally thought and the observed lack of ageing in some species highlights the importance of trade-offs in life history evolution.

## Methods

Detailed exposition and derivations are in the Supplementary Methods.

**Demography, selection, and genetic effects**. We consider a demographically stable population of constant, essentially infinite size where $m_j$ is average fecundity, i.e., number of offspring, at individual age $j$ and $p_j$ is average survival, i.e., fraction surviving, from age $j$ to age $j + 1$. Reproduction starts at age $\alpha$ and continues indefinitely. We define

- ageing in fecundity as $m_j > m_{j+1}$ with $j = \alpha, \alpha + 1, \ldots$
- negative ageing in fecundity as $m_j < m_{j+1}$ with $j = \alpha, \alpha + 1, \ldots$
- ageing in survival as $p_j > p_{j+1}$ with $j = \alpha - 1, \alpha, \ldots$
- negative ageing in survival as $p_j < p_{j+1}$ with $j = \alpha - 1, \alpha, \ldots$

The mean fitness in the population is

$$\bar{w} = \sum_{i=1}^{\infty} m_i p_0 p_1 \cdots p_{i-1}, \tag{1}$$

which can be seen as the reproductive value at birth in the demographically stationary state[15,51,52]. Mean fitness in the neutral population is 1. In this setting, the classic theory[8] computes the selection force on age-specific fecundity and survival as proportional to the gradients

$$\frac{\partial \bar{w}}{\partial m_j} = p_0 p_1 \cdots p_{j-1} \quad \text{and} \tag{2a}$$

$$\frac{\partial \bar{w}}{\partial \ln p_j} = \sum_{i=j+1}^{\infty} m_i p_0 p_1 \cdots p_{i-1}, \tag{2b}$$

respectively, which are always positive, as fecundity and survival are direct fitness components. Supposing fecundity does not cease and survival is never perfect ($<1$) at any age,

$$\frac{\partial \bar{w}}{\partial m_{j+1}} < \frac{\partial \bar{w}}{\partial m_j} \quad \text{and,} \tag{3a}$$

$$\frac{\partial \bar{w}}{\partial \ln p_{j+1}} < \frac{\partial \bar{w}}{\partial \ln p_j}, \tag{3b}$$

which are the classic theory result[8] of the steady decline of these gradients with age. Equation 2 presupposes that genetic effects are additive on fecundity, whence the identity function implicitly operating on $m_j$ on the left-hand side of eq. 2a, and proportional on survival, whence the natural logarithm operating on $p_j$ on the left-hand side of eq. 2b. To make this explicit, we introduce the functions $f_M$ and $f_P$, which act on age-specific fecundity and on age-specific survival, respectively. We only assume these functions be strictly increasing, to preserve positive selection on $f_M(m_j)$ and $f_P(p_j)$, and twice differentiable. We use them to add appropriate weights to the classic selection gradients and get the general gradients,

$$\frac{\partial \bar{w}}{\partial f_M(m_j)} = \frac{1}{f'_M(m_j)} \frac{\partial \bar{w}}{\partial m_j} \quad \text{and,} \tag{4a}$$

$$\frac{\partial \bar{w}}{\partial f_P(p_j)} = \frac{1}{p_j f'_P(p_j)} \frac{\partial \bar{w}}{\partial \ln p_j}, \tag{4b}$$

which, as expected[13], show no obvious age pattern. As an example of usage of these gradients, we recover the classic theory of additive effects on fecundity and proportional effects on survival when $f_M(m_j) = m_j$ and $f_P(p_j) = \ln p_j$. We can reverse the classic assumptions, as in Fig. 1, by setting $f_M(m_j) = \ln m_j$ and $f_P(p_j) = p_j$. But the minimal assumptions about the $f$ functions allow virtually any form of genetic effects, and not only additive and proportional, on fecundity and survival. For example, since $-\ln p_j$ is average mortality between ages $j$ and $j + 1$, when $f_P(p_j) = -\ln(-\ln p_j)$ we have proportional decrements in mortality. We emphasize that the $f$ are not fitness functions. They only capture how age-specific genetic variation acts on fecundity and survival.

**Model**. To build our dynamical model, we use the breeder's equation from quantitative genetics[20],

$$\bar{z}_j(t+1) = \bar{z}_j(t) + g_j \frac{\partial \bar{w}}{\partial \bar{z}_j}(t) + \sum_{i \neq j} g_{ji} \frac{\partial \bar{w}}{\partial \bar{z}_i}(t), \tag{5}$$

which describes the time-discrete evolution of a mean trait ($\bar{z}_j$) in a very large population under the action of selection when other traits ($\bar{z}_i$ with $i \neq j$) are also under selection. In this equation, the change in a focal trait over one-time step is the sum of two terms. The first term is the product between the selection gradient $\partial \bar{w}/\partial \bar{z}_j$ and the additive genetic variance $g_j$ for that trait. This term captures the role of the selection force alone on the trait, provided there is some variability for it ($g_j > 0$). The second term, the sum in eq. (5), captures trade-offs between the focal trait and all other traits concomitantly subject to selection with $g_{ji}$ the genetic covariance between traits $i$ and $j$. In our usage of the breeder's equation, we set

covariances to zero so that trade-offs are absent. The additive genetic variance, the fuel of selection, is assumed very small, constant over time and equal for all traits, $g_j = \delta$ with $0 < \delta \ll 1$. Thus, all traits share and retain, the same potential to evolve and selection is weak. These assumptions are also implicit in the classical work by Hamilton[8]. We study fecundity evolution and survival evolution separately.

When we let fecundity evolve, survival at each age is kept a positive constant at all times but it may vary between ages. We set $\bar{z}_j = f_M(m_j)$ and use the selection gradient in eq. 4a inside the breeder's equation. We then track change over time at the level of each $m_j$ via

$$m_j(t+1) = \Omega_M(t) f_M^{-1}\left( f_M(m_j(t)) + \delta \frac{1}{T(t) f_M'(m_j(t))} \frac{\partial \bar{w}}{\partial m_j}(t) \right), \qquad (6)$$

where $f_M^{-1}$ is the inverse function of $f_M$ and the quantity $T$ is the average generation time, which is required to get the change per time step from the change $\delta \frac{1}{f_M'(m_j)} \frac{\partial \bar{w}}{\partial m_j}$ per generation. The factor $\Omega_M(t)$ describes density dependence. We assume a constant population size regulated by ecological factors, which is achieved by scaling fecundity at all ages equally to ensure that $\bar{w}(t) = \bar{w}(t+1) = 1$, as suggested before[53].

When we look at how survival evolves, we set $\bar{z}_j = f_P(p_j)$ and use the selection gradient in eq. 4b inside the breeder's equation. Fecundities are set to positive constant parameters independent of one another. We then track change over time at the level of each $p_j$ via

$$p_j(t+1) = \Omega_P(t) f_P^{-1}\left( f_P(p_j(t)) + \delta \frac{1}{T(t) p_j f_P'(p_j(t))} \frac{\partial \bar{w}}{\partial \ln p_j}(t) \right). \qquad (7)$$

Considerations analogous to eq. (6) apply.

As we presuppose potentially infinite ages, eqs. (6) and (7) give each rise to a separate dynamical system in the space of non-negative sequences of real numbers. The metric we use for this space is

$$d_i((a_n), (b_n)) = \sum_{j=i}^{\infty} \frac{1}{2^j} \frac{|a_j - b_j|}{1 + |a_j - b_j|}, \qquad (8)$$

with $(a_n)$ and $(b_n)$ sequences and starting index $i$ depending on the specific tail sequence of interest. The distance between $(a_n)$ and a subset $B$ of the space is $d_i((a_n), B) = \inf_{(b_n) \in B} d_i((a_n), (b_n))$.

**Analysis of fecundity.** We change coordinates for the system in eq. (6) to $y_j = m_{j+1}/m_j$. With a first-order Taylor expansion around $\delta = 0$, the transformed dynamics of fecundity are,

$$y_j(t+1) - y_j(t) = M_j(t) \left[ \left( \frac{\partial \bar{w}}{\partial m_{j+1}}(t) \middle/ \frac{\partial \bar{w}}{\partial m_j}(t) \right) - \frac{m_{j+1}(t) f_M'(m_{j+1}(t))^2}{m_j(t) f_M'(m_j(t))^2} \right], \qquad (9)$$

where the factor $M_j$ remains positive at all times. Setting to zero the left-hand side of eq. (9) to get equilibria (denoted by asterisks), we find that, for ages $j = \alpha, \alpha+1, \ldots$,

$$\frac{p_0 p_1 \cdots p_j}{f_M'(m_{j+1}^*)} = \frac{p_0 p_1 \cdots p_{j-1}}{f_M'(m_j^*)} \sqrt{p_j \frac{m_{j+1}^*}{m_j^*}}. \qquad (10)$$

From eqs. 2a and 4a we see in eq. (10) that the equilibrium selection gradient on fecundity declines with age when ageing in fecundity is an equilibrium. Independently of our choice of $f_M$, there is no realistic equilibrium of the form $m_j^* = m_{j+1}^*$ with $j = \alpha, \alpha+1, \ldots$, as this would lead to $p_j = 1$ for $j = \alpha, \alpha+1, \ldots$ implying the total absence of mortality during adulthood. Depending on the sign of $\partial(m_i f_M'(m_i)^2)/\partial m_i$, equilibrium fecundity, if it exists, shows ageing or negative ageing (Fig. 2). When $\partial(m_i f_M'(m_i)^2)/\partial m_i > 0$, ageing in fecundity is an equilibrium ($y_j^* < 1$ with $j = \alpha, \alpha+1, \ldots$). The distance between the evolving fecundity schedule $(y_n) = y_\alpha, y_{\alpha+1}, \ldots$ and the set $A_<^m$ of life histories with ageing in fecundity is

$$d_\alpha((y_n), A_<^m) = \sum_{i=\alpha}^{\infty} \frac{\Theta(y_i - 1)}{2^i} \frac{|y_i - 1|}{1 + |y_i - 1|}, \qquad (11)$$

where $\Theta$ is the unit step function. This distance can be shown to decrease under eq. (9) when $\partial(m_i f_M'(m_i)^2)/\partial m_i > 0$. In particular, when $f_M(m_i) = m_i^q$ with $q > 1/2$,

$$V^m = \sum_{i=\alpha}^{\infty} \frac{1}{2^i} \frac{\left( p_i - y_i^{2q-1} \right)^2}{1 + \left( p_i - y_i^{2q-1} \right)^2} \qquad (12)$$

is a Lyapunov function and when $f_M(m_i) = a_1 \exp(cm_i)/c + a_2$ with $a_1 > 0$ and $c > 0$, any equilibrium can be shown linearly stable. When $\partial(m_i f_M'(m_i)^2)/\partial m_i < 0$, negative ageing in fecundity may be an equilibrium ($m_{j+1}^* > m_j^*$ with $j = \alpha, \alpha+1, \ldots$). Therefore, $y_j^* > 1$ with $j = \alpha, \alpha+1, \ldots$. The distance between $(y_n)$ and the set of $A_>^m$ of life histories with negative ageing in fecundity is

$$d_\alpha((y_n), A_>^m) = \sum_{i=\alpha}^{\infty} \frac{\Theta(-(y_i - 1))}{2^i} \frac{|y_i - 1|}{1 + |y_i - 1|}. \qquad (13)$$

This distance can be shown to increase with time under eq. (9) when

$\partial(m_i f_M'(m_i)^2)/\partial m_i < 0$ and $(y_n)$ is not in $A_>^m$. Moreover, keeping $m_\alpha$ at equilibrium value and looking at the system in eq. (6) after a first-order Taylor expansion about $\delta = 0$, it can be shown that the distance

$$d_{\alpha+1}((m_n(t))_{\alpha+1}, (m_n^*)_{\alpha+1}) = \sum_{j=\alpha+1}^{\infty} \frac{1}{2^j} \frac{|m_j(t) - m_j^*|}{1 + |m_j(t) - m_j^*|} \qquad (14)$$

between the subsequence $(m_n)_{\alpha+1} = m_{\alpha+1}, m_{\alpha+2}, \ldots$ and its equilibrium, if this exists, increases with time when $\partial(m_i f_M'(m_i)^2)/\partial m_i < 0$. Irrespective of the sign of $\partial(m_i f_M'(m_i)^2)/\partial m_i$, it can be shown from eq. (10) that at most a single equilibrium with either ageing or negative ageing is possible—otherwise a contradiction with demographic stationarity is obtained.

**Analysis of survival.** We change coordinates for the system in eq. (7) to $x_j = p_{j+1}/p_j$. With a first-order Taylor expansion around $\delta = 0$, the transformed dynamics of survival are

$$x_j(t+1) - x_j(t) = P_j(t) \left[ \left( \frac{\partial \bar{w}}{\partial \ln p_{j+1}}(t) \middle/ \frac{\partial \bar{w}}{\partial \ln p_j}(t) \right) - \frac{p_{j+1}^2(t) f_P'(p_{j+1}(t))^2}{p_j^2(t) f_P'(p_j(t))^2} \right], \qquad (15)$$

where the factor $P_j$ remains positive at all times. Setting the left-hand side of this expression to zero to calculate equilibria, we find that, for ages $j = \alpha-1, \alpha, \ldots$,

$$\frac{1}{p_j^* f_P'(p_j^*)} \left( \frac{\partial \bar{w}}{\partial \ln p_j} \bigg|_{(p_n)=(p_n^*)} \right)^{\frac{1}{2}} = \frac{1}{p_{j+1}^* f_P'(p_{j+1}^*)} \left( \frac{\partial \bar{w}}{\partial \ln p_{j+1}} \bigg|_{(p_n)=(p_n^*)} \right)^{\frac{1}{2}}, \qquad (16)$$

which, by eqs. 2b and 4b, implies that the equilibrium selection gradient always declines with reproductive age. Independently of our choice of $f_P$, there is no equilibrium of the form $p_j^* = p_{j+1}^* > 0$ with $j = \alpha-1, \alpha, \ldots$, as substituting this into eq. (16) and using eq. 2b we would get $m_j p_0^* p_1^* \ldots p_{j-1}^* = 0$ for $j = \alpha, \alpha+1, \ldots$. The same problem derives from the assumption that $f_P$ is the natural logarithm. Since this is what the classic theory assumes, this theory fails to have an equilibrium in our model. Depending on the sign of $\partial(p_i f_P'(p_i))/\partial p_i$, equilibrium survival, if it exists, shows ageing or negative ageing (Fig. 3). When $\partial(p_i f_P'(p_i))/\partial p_i > 0$, only ageing in survival may be an equilibrium ($x_j^* < 1$ with $j = \alpha-1, \alpha, \ldots$). The distance between the evolving survival schedule $(x_n)_{\alpha-1} = x_{\alpha-1}, x_\alpha, \ldots$ into reproductive ages and the set $A_<^p$ of life histories with ageing in survival is

$$d_{\alpha-1}((x_n)_{\alpha-1}, A_<^p) = \sum_{i=\alpha-1}^{\infty} \frac{\Theta(x_i - 1)}{2^i} \frac{|x_i - 1|}{1 + |x_i - 1|}. \qquad (17)$$

It can be shown that this distance tends to zero with time under eq. (15) when $\partial(p_i f_P'(p_i))/\partial p_i > 0$. When $\partial(p_i f_P'(p_i))/\partial p_i < 0$, negative ageing in survival may be an equilibrium ($x_j^* > 1$ with $j = \alpha-1, \alpha, \ldots$). The distance between the set $A_>^p$ of life histories with negative ageing in survival and an evolving survival schedule outside of this set is

$$d_{\alpha-1}((x_n)_{\alpha-1}, A_>^p) = \sum_{i=\alpha-1}^{\infty} \frac{\Theta(-(x_i - 1))}{2^i} \frac{|x_i - 1|}{1 + |x_i - 1|}. \qquad (18)$$

This distance can be shown to increase with time under eq. (15) when $\partial(p_i f_P'(p_i))/\partial p_i < 0$.

**Reporting summary.** Further information on research design is available in the Nature Research Reporting Summary linked to this article.

## Data availability
The authors declare that the data supporting the findings of this study are available within this paper and its supplementary information files.

## Code availability
Supplementary Code 1 includes commented R code to generate and plot the data for Fig. 1 in the main text and figures in the supplementary information file.

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

## Acknowledgements

The authors thank Annette Baudisch for comments on an earlier draft and the Max Planck Society for funding.

## Author contributions

S.G. conceived, developed, analyzed the model, and wrote a first draft of the manuscript. S.G. and A.T. reviewed the analysis and wrote the manuscript.

## Funding

## Competing interests

The authors declare no competing interests
