## [Peer Review File · Nature Communications]

The selection force weakens with age because ageing evolves and not vice versaReviewers' Comments:

Reviewer #1:

Remarks to the Author:

This paper combines familiar elements of evolutionary models of ageing in an intriguingly novel way, and so has the potential to open up new conceptual approaches to the subject. In particular, the observation that the logarithmic link is not simply a convenient stand-in for all possible link functions, but may actually represent an extreme that behaves qualitatively differently from any plausible alternative should inspire a lot of reconsideration of past assumptions. The emphasis on stability analysis is welcome, and some of the technical arguments are ingenious.

I have some doubts that the current format serves this paper well. In order to fit the journal's requirements, what is essentially a 50+ page mathematical biology paper has been split into a 10-page journal article with a dense 40-page "supplement". The supplement does not, in my opinion, allow itself to be easily severed from the rest, and I don't think the main paper can be well understood without at least some of the technical detail of the supplement.

In particular, the role of the functions f_P and f_M in the dynamics were, for me, impossible to decipher from the main paper, though it became clear — and I quite appreciated the idea! — after some engagement with the supplement.

The paper cites relatively little other mathematical work on evolution of ageing, other than Hamilton's original work the papers of A Baudisch and collaborators. A series of papers and a monograph by Wachter, Evans, and Steinsaltz address precisely the problem of generalised interactions between sites, and consider the dynamical implications. This model and its constraints may be different in important ways to those of the present paper, but it would be useful to at least discuss the ways the two approaches overlap.

One important technical matter that may be a crucial error, or may represent some confusion on my part: Quite a lot of what follows depends on equation SI.31. But I don't see where that comes from. As I understand it, we are looking for a solution that makes the coefficient of δ in SI.30 equal to 0. But if all of the terms $H_j/p_j^2 f'(p_j)^2$ are equal to the same constant A , then that coefficient becomes $A(1 - \sum H_j^*)$, and I don't think there's any reason why that sum should be equal to 1. If this is indeed correct, I think some additional explanation is required.

David Steinsaltz

Reviewer #2:

Remarks to the Author:

This paper is written in quite an antagonizing way: it makes claims about previous literature having been operating on an ad hoc assumption basis, and promises to rectify the situation.

The problem is that I don't think the criticism of earlier literature is really valid. My complaint here takes two forms: first, are the authors being uncharitable towards the cited work? I believe, sadly, yes. Second, do they complain about the state of the literature as a whole based on only citing a subset of it? Again, unfortunately, yes. See comments 1 and 2 below, before I move on to the details of the model.

1. The central claim here boils down to a gist "others have just assumed that the force of selection declines with age". Is this true? Hamilton 1966, the classic that's particularly being complained about here, writes towards the end of his paper "It is striking to find that even under these utopian conditions selection is still so orientated that, given genetical variation, phenomena of senescence will

tend to creep in.” which to me is a very clear indication that he did try to break the pattern as hard as he could, and still it was there. This is not just an impression from the verbal descriptions of what he did; it is also corroborated by the maths. It would be weird indeed if he’d assumed the thing he wants to find out.

2. Regarding my ‘subset’ complaint, it is interesting that the authors contrast their findings which contain an assumption of constant population size (very strict density dependence) to earlier work that largely ignored density dependence. They do so without acknowledging the lively debate of the way density dependence can alter predictions on senescence: see e.g. the debates between Moorad et al. and Day & Abrams in TREE (and some others).

3. Mathematically, the worrying aspect of the present model is that there is a deviation from the classic assumption that a completely non-senescent organism should have an exponentially distributed lifespan. Instead, the authors assume a maximum age beyond which survival is impossible. That to me is actually an a priori assumption of ageing; so I spent quite some time looking for what the authors here even consider as their operational definition of ageing, and eventually found it in the supplementary material. (A side issue: for the cognitive load placed on a reader, not having a clear definition in the main text non-ideal.) This makes me doubt the bold claims made as a whole. The issue is that they actually assume, a priori, that ageing occurs, if we define ageing in a broad sense (here, no survival chances at all beyond a specific age); the question then becomes: assuming this ‘unavoidable’ part of ageing is there, does this have an impact on that part of ageing that is free to evolve and that is said to occur according to definitions in the supplementary section 1.2? This way of phrasing the question reveals what might be driving much of their results: an old organism is a priori forced to have only little of its life left. Importantly, this limit is not present in the case in a memoryless process of exponentially distributed lifespans (the reason why Hamilton considers integrals and sums that go all the way to infinity).

So, to me, the authors seem to have shown that certain results differ from classical theory because of a combination of having chosen a life history with a finite lifespan (thus they presuppose ageing in reality, although not necessarily using their narrow definition given in the supplementary), and constrain their view to one specific form of density regulation. They don’t write about these effects at all, however, perhaps not realizing that these choices can be important; they instead write in a very feisty ‘grandspeak’-like manner accusing others of assuming what was to be proven – when they in reality haven’t done so. I believe, therefore, that they may have misidentified the reasons behind any discrepancies.

Finally, some comments about the quality of writing.

4. I must say that despite a very extensive and thorough supplementary material, the paper as a whole is not clearly written. The schematic view provided in Figure 2 remains largely a mystery to me; I can see (and understand) 3 different regions in each oval but I don’t know what the vertical dimension is meant to represent, or what happens (= what changes are observed in the dynamics) if one moves horizontally while remaining in the no-ageing region; I also don’t know what the big blob in the top right figure signifies. I know this figure is meant to be schematic/conceptual only, but it should ideally help to understand the issues, not cause further bafflement.

5. An unfortunate feature of the MS is that the key result is derived using functions f which are themselves not explained very well; when we’re told that they are ‘generic functions that capture any possible mutational effect on survival and fecundity at each age’, then I can see what is being said on line 112, but the following claim I do not. I reread Hamilton twice now and still I do not understand where Hamilton is assuming that f_p is the natural logarithm. Of course, one shouldn’t refrain from criticising classics, but when making bold claims it should be easy to see what one is talking about (it doesn’t help that f in Hamilton’s notation differs from the present authors’ choice).

6. There are also numerous claims that I only understood once I took an "OK, if you want to express it that way, then I guess technically you can" approach - when ideally it should have been clear immediately. For example, they rather weirdly appear to complain (lines 159-160) about the classic model having an age-specific effect limited to a single fitness component, only to admit that they themselves make the same assumption. This makes me feel like countering that Hamilton did include a model where several ages (from a particular age onwards) experience elevated mortality. Thereafter, my "OK I guess this is OK after all..." moment was about realizing that "age-specific effects" can technically be interpreted broadly to include "from age i onwards" type effects. But I hope you see where I'm coming from - a more natural interpretation of a statement like that is to assume you're talking about an effect on vital rates that occurs at one age and only that one age.

Reviewer #3:

Remarks to the Author:

In this manuscript the authors propose a dynamic evolutionary model that has generic functions, f sub P and f sub M, capture the way mutations affect survival and fecundity. They find that in general, not just for the assumptions made in the classic approach towards the evolution of aging, declining force of selection (in the classic setting) is a result of aging rather than the necessary condition for aging to evolve.

This is a lucid, clear, and in particular well-written manuscript. One can see that the authors have not only spent a lot of thought on the issue, but have also spent great effort on presentation and visualization. The language is crystal clear. The manuscript makes the reader re-think ingrained assumptions about the evolutionary theory of aging - a much needed approach.

My comments are minor - three are relatively major but do not as such diminish the validity of this work.

Major minor points:

1. It is in fact well known that the way the force of selection declines over ages is a function of aging (as well as the driving force of aging in the classic approach). Thus, while I appreciate the novelty of the model and the generality of f sub P and f sub M, the presentation could be tuned down a little bit on the decline of the force of selection being a result of aging. To be clear, I do not think that the authors wish to sweep this fact under the rug; rather, it is a matter of emphasis, of presentation. And indeed I agree that this is something that the classic theories have never really dealt with: if there is a positive feedback loop, then why does the rate of aging, however defined, not escalate without bound?
2. Similarly, it is already known that different assumptions about the way mutations affect survival and fecundity lead different conclusions (refs 10 and 27 of the manuscript, for example). Thus, what the authors add with their manuscript is the very nice dynamic picture of it all. It would be good if this were fleshed out a little more in the discussion.
3. There has been some recent work by Levin and Levy, *The Biostatistics of Aging: From Gompertzian Mortality to an Index of Aging-Relatedness*, that also tries to overcome some of the limitations of the classic approach that the authors of the current manuscript criticize. In particular, they apply the causal pie model of causation (typing "causal pie model aging" in Google Scholar gives quite some results, in case the authors are not familiar with this concept) to investigate how the interaction of small changes affects the picture (objecting to this idea of small additive changes that underlies the classic approach). The authors might find it interesting to discuss this in regards to their work.
4. I do not think that the authors follow entirely through on their own argument. Regarding observed patterns in nature, in their (very brief) discussion, the authors state: "Ageing is only one of the many patterns that these trajectories can follow." (lines 141-142). Then they remind the reader that their own results give aging as the only stable endpoint (lines 156-157). They then conclude, without much qualification, that trade-offs must be the missing part of the puzzle. I understand that this is convenient, the G matrix could do this, as the authors point out (although in their model it is a scalar multiple of the identity), but some might say that the latter is at least as much an ad-hoc assumption

as Hamilton's descending selection gradients! Why trade-offs? Why not other constraints? Indeed, the authors consider only iteroparity and a constant G matrix, while the discussion of the manuscript seems to take a more general view. And evolution is never finished, so would the authors not be as bold as to speculate that the non-aging species currently observed are transient, at least for iteroparous species?

Minor minor points:

1. Lines 144-146: "This is questionable, as the same theoretical argument that is made to derive this principle leads to a contradiction of it under slightly different assumptions." Not sure what is meant here. What is the contradiction? Reference 10 given, Baudisch PNAS 2005, calculates selection gradients for non-additive perturbations. Reference 27, also from Baudisch' group, shows that the same can be achieved by making the perturbations (rather than the selection gradients) proportional to mortality, after which the product of the two is integrated out to give the fitness effect. So it seems to be more about the biological argument rather than the choice of mathematical representation of the biologics: biologics, not mathematics, make aging inevitable or not. Might this be what the authors mean? (MJ Wensink, TF Wrycza, A Baudisch - PloS one, 2014 seems to make an argument somewhat akin)
2. Caswell proved that Hamilton's selection gradients can be written as the product of reproductive value and the stable age distribution. Since the authors introduce these concepts in the appendix, it might be nice to remark on this there. I think the reference is https://www.jstor.org/stable/26349604?seq=1#metadata_info_tab_contents (but please check).
3. Probably not and I haven't thought this through, but could the generality of $f_{sub P}$ and $f_{sub M}$ confound with the variance of the G matrix in predicting the evolutionary dynamics, at least in the short term, if G evolves? There seem to be unlimited degrees of freedom for both, so I suspect there might be some cancelling out to do, at least by approximation, taking first Taylor terms, something like that.

(The appendix is a small book – I hope that I am forgiven for not checking the equations line by line, but it certainly looks tidy and considered!)

Reply to Reviewers

We are grateful to the Reviewers for their careful analysis of our manuscript and for bringing to our attention a number of issues. Your comments were very helpful for us to identify problematic points that we had overlooked. Tackling these points, we have written an entirely new manuscript that we are now re-submitting to Nature Communications. The new version, we believe, improves and expands upon the initial version of our work by addressing all points that were made. For a point-by-point reply to your comments and on how we have dealt with them please see below.

Reply to Reviewer 1

This paper combines familiar elements of evolutionary models of ageing in an intriguingly novel way, and so has the potential to open up new conceptual approaches to the subject. In particular, the observation that the logarithmic link is not simply a convenient stand-in for all possible link functions, but may actually represent an extreme that behaves qualitatively differently from any plausible alternative should inspire a lot of reconsideration of past assumptions. The emphasis on stability analysis is welcome, and some of the technical arguments are ingenious.

Thanks for your accurate and very positive judgement of our submission and for finding merit to it!

1. COMMENT: I have some doubts that the current format serves this paper well. In order to fit the journal's requirements, what is essentially a 50+ page mathematical biology paper has been split into a 10-page journal article with a dense 40-page "supplement". The supplement does not, in my opinion, allow itself to be easily severed from the rest, and I don't think the main paper can be well understood without at least some of the technical detail of the supplement.

REPLY: For this resubmission, we have opted for a different format in which the technical part is explained for one specific case.

2. COMMENT: In particular, the role of the functions f_P and f_M in the dynamics were, for me, impossible to decipher from the main paper, though it became clear — and I quite appreciated the idea! — after some engagement with the supplement.

REPLY: We have re-written the main text. In particular, we have introduced more properly f_P and f_M inside the new Methods Section, absent in the original submission, that condensates all our assumptions and the key passages to our results.

3. COMMENT: The paper cites relatively little other mathematical work on evolution of ageing, other than Hamilton's original work the papers of A Baudisch and collaborators. A series of papers and a monograph by Wachter, Evans, and Steinsaltz address precisely the problem of generalised interactions between sites, and consider the dynamical implications. This model and its constraints may be different in important ways to those of the present paper, but it would be useful to at least discuss the ways the two approaches overlap.

REPLY: Yes, we agree with Reviewer 1 that we did not elaborate enough upon the connections with other work in the field. In the resubmission, we have expanded upon the Discussion Section to bridge our results with published ones including the series of work of Wachter, Evans, and Steinsaltz on the topic of evolution of ageing. However, we mostly note the difficulties in comparing their work with ours.

4. COMMENT: One important technical matter that may be a crucial error, or may represent some confusion on my part: Quite a lot of what follows depends on equation SI.31. But I don't see where that comes from. As I understand it, we are looking for a solution that makes the coefficient of δ in SI.30 equal to 0. But if all of the terms $H_j/p_j^2 f'(p_j)^2$ are equal to the same constant A , then that coefficient becomes $A(1 - \sum H_j^*)$, and I don't think there's any reason why that sum should be equal to 1. If this is indeed correct, I think some additional explanation is required.

REPLY: Unfortunately, we failed to mention this in the Supporting Information, but the sum of Hamilton's gradients over age always is 1, indeed. Starting from

$$H_j = \frac{1}{T} \sum_{i=j+1}^{\infty} l_i m_i \lambda^{-i}, \quad (1)$$

we have that

$$\sum_{j=0}^{\infty} H_j = \frac{1}{T} \sum_{j=0}^{\infty} \sum_{i=j+1}^{\infty} l_i m_i \lambda^{-i} = \frac{1}{T} \sum_{i=1}^{\infty} i l_i m_i \lambda^{-i} = \frac{1}{T} T = 1 \quad (2)$$

where the double series is reduced to one by swapping from row sums to column sums in the following infinite table:

Age 1	Age 2	Age 3	Age 4	...	Age n	...
$l_1 m_1 \lambda^{-1}$	$l_2 m_2 \lambda^{-2}$	$l_3 m_3 \lambda^{-3}$	$l_4 m_4 \lambda^{-4}$...	$l_n m_n \lambda^{-n}$...
	$l_2 m_2 \lambda^{-2}$	$l_3 m_3 \lambda^{-3}$	$l_4 m_4 \lambda^{-4}$...	$l_n m_n \lambda^{-n}$...
		$l_3 m_3 \lambda^{-3}$	$l_4 m_4 \lambda^{-4}$...	$l_n m_n \lambda^{-n}$...
			$l_4 m_4 \lambda^{-4}$...	$l_n m_n \lambda^{-n}$...
				\ddots	\vdots	...
					$l_n m_n \lambda^{-n}$...

In the new version of the Supporting Information, we have made this clear. Thanks for noting this omission from our part! Please also note that the sums become series in the re-submission, as we were prompted by another Reviewer to remove the assumption of an arbitrarily large, yet finite number of ages. Our previous results transfer smoothly to the infinite dimensional case, although techniques are now different in several aspects.

Reply to Reviewer 2

This paper is written in quite an antagonizing way: it makes claims about previous literature having been operating on an ad hoc assumption basis, and promises to rectify the situation. The problem is that I don't think the criticism of earlier literature is really valid. My complaint here takes two forms: first, are the authors being uncharitable towards the cited work? I believe, sadly, yes. Second, do they complain about the state of the literature as a whole based on only citing a subset of it? Again, unfortunately, yes. See comments 1 and 2 below, before I move on to the details of the model.

Thanks for engaging with our manuscript and urging us to revisit our results and writing style.

1. COMMENT: The central claim here boils down to a gist “others have just assumed that the force of selection declines with age.” Is this true? Hamilton 1966, the classic that's particularly being complained about here, writes towards the end of his paper “It is striking to find that even under these utopian conditions selection is still so orientated that, given genetical variation, phenomena of senescence will tend to creep in.” which to me is a very clear indication that he did try to break the pattern as hard as he could, and still it was there. This is not just an impression from the verbal descriptions of what he did; it is also corroborated by the maths. It would be weird indeed if he'd assumed the thing he wants to find out.

REPLY: We understand from this comment that Reviewer 2 finds Hamilton's arguments about the inevitable decline in the selection force persuasive. We agree with Reviewer 2 that it may sound weird what is stated in the introduction of our work in this respect that Hamilton had somewhat implicitly assumed what he wanted to find out. However, apparently this is precisely what he did, most probably unknowingly. We would also assume that Hamilton tried hard to break the pattern, but probably not in the way that Annette Baudisch proposed 40 years later: That Hamilton made an assumption that was necessarily conducive to a declining force of selection with age was first shown by Baudisch (2005), who is not an author of the submitted manuscript. Thus, we cannot claim absolutely any credit for showing this, as Reviewer 2 appears to suggest. Annette Baudisch did not question Hamilton's mathematics (nor we do). Given Hamilton's assumptions, the mathematics thereafter is just fine and highly influential. But Baudisch did question one of his assumptions: proportional mutational effects on survival. And she found that equally valid, alternative assumptions do not necessarily lead to a decline in the selection force. Her result has since been hailed as a major insight over the last 35 years, see Flatt and Partridge (2018), as we also stated in lines 26-30 and refs. 10-2 of the original submission. This is where we start from: Given the important discovery (by others!) that there are cases in which the selection force may not decline with age, what happens in the long run if we let evolution operate when just any mutational effect is allowed? This is the question we wanted to answer in our work. Admittedly, in answering this question, we agree with Baudisch's criticism of Hamilton's somewhat restricted perspective on the selection force.

We are then puzzled by Reviewer 2's comment that this would imply our being uncharitable towards Hamilton and we are unsure what we can do about this other than rephrasing the introduction, as we do in the new submission, so that we put even more emphasis on the credit due to Baudisch for her critique of Hamilton. But we are entirely open to specific indications by Reviewer 2 on whether and how Baudisch's results about the selection force should be rejected and those of Hamilton rehabilitated. Finally, we should also underline that our results turn out to eventually rescue Hamilton's own conclusion about the inevitability of ageing and the decline in the selection force, although via a completely different argument from his. In this way, we suggest a solution as to how the problem raised by Baudisch can be overcome. We stated this in lines 147-56 of the original submissions. These lines ended with “our work shows that the [assumed decline in the selection force], once solved, does not undermine the explanatory power of the classic model.” This further adds to our puzzlement as to the alleged uncharitable treatment of Hamilton's work that Reviewer 2 finds in our work.

2. COMMENT. Regarding my ‘subset’ complaint, it is interesting that the authors contrast their findings which contain an assumption of constant population size (very strict density dependence) to earlier work that largely ignored density dependence. They do so without acknowledging the lively debate of

the way density dependence can alter predictions on senescence: see e.g. the debates between Moorad et al. and Day & Abrams in TREE (and some others).

REPLY: Thanks for bringing this issue up. We have included in this re-submission references to the mentioned debate and we have explained the role of density dependence in our model in more detail. Four things should be noted in this respect.

- (a) First, as Reviewer 2 notes, “earlier work [on senescence] largely ignored density dependence.” Indeed, they ignored it largely – but not entirely. Hamilton’s classic 1966 paper is not concerned with the density independent case only. In his 1966 paper, he included a discussion (p. 36) of how to merge his view of the evolution of the mortality trajectory with a mechanism of density dependence which is precisely along the lines of ours: a drop of equal magnitude in mortality at all ages. Speaking about his own paper, Hamilton himself would later on comment that “Williams, on the other hand, seemed to have understood the matter right through and had merely sacrificed a little of the generality that I had attained by assuming the populations under selection to be static. But when all work for my paper was done, I had to admit that even this defect was trivial because most populations had to be almost static in the long term. In short, it needs a very peculiar species and a very unstable ecology to make the elegant Eulerian weightings in the general version of the theory really necessary” (Hamilton, 1996, p. 89). In summary, Hamilton regarded the density dependent case as the most realistic. For historical accuracy, we should also add that both Medawar (1958) and Williams (1957) in their classic works on senescence assume stationary populations and therefore some form of density dependence.
- (b) The debate Reviewer 2 is mentioning adheres to Hamilton’s assumption of additive genetic changes in mortality (proportional genetic changes in survival). Therefore, it cannot be directly informative about our study of generalized genetic effects on mortality changes that heavily depart from this assumption. Moreover, the debate is not about density dependence only. The debate is about the combined effect of different forms of density dependence and the level of age-independent mortality. The debate is not at all concerned with how these different combinations (the only part that varies is the density dependence mechanism) may or may not impact on whether senescence or its opposite (negative senescence or negligible senescence) evolve or whether the selection gradient is always declining (these are our problems of interest, instead). The crux of that debate is instead on whether senescence (assumed to evolve along the lines of Hamilton’s original argument) should be more or less fast, i.e., how fast mortality increases with age and/or fecundity declines with age, for the given combination of density dependence mechanism and the given level of age-independent mortality. The pattern of ageing and the pattern of the selection gradient are not put into question, they are both assumed declining functions of age and they are not modelled as the result of a long term evolutionary process under the studied combination of density dependence and age-independent mortality. In our work, we do not assume any such pattern. We also ask a very different question: What is the effect of selection on any initial life history under any mutational effect? We find that the long run tendency in evolution is the emergence of ageing: a declining fertility and increasing mortality with age.
- (c) Another reason that the mentioned debate cannot be directly informative about the role played by density dependence in our work is that this entire debate examines the combined effect of density dependence and age-independent mortality with the important qualification that the latter is extrinsic, in the sense that it is due to “mortality factors that are independent of age and condition” (Abrams, 1993, p. 878) so that no mutant can ever escape this form of mortality. As Abrams (1993, p. 883) notes in his paper, from which the entire debate originates, inside the Section ‘Nonextrinsic Mortality’: “there has never been any demonstration that a major mortality factor of a natural population is completely independent of age and state of senescence. If most mortality is nonextrinsic, then correlations between observed mortality rates and rates of senescence may not reflect any of the above theoretical results.” And we add, by extension, the results of the whole debate on extrinsic mortality do not directly bear on the case of nonextrinsic mortality. In our model, we cannot separate mortality into extrinsic and nonextrinsic factors. The density dependent mechanism in our model for survival evolution reduces survival at all ages by the same

proportion, the selection gradient is computed on the whole resulting survival in the next time step (without accounting for any source of extrinsic mortality) and the life history then moves in the direction of greatest fitness increase according to the breeder's equation.

- (d) We aim to have a long term view on life history evolution. Therefore, we cannot but introduce a form of density dependence. Selection for survival and fecundity is always positive. In the absence of any ecological pressure, the population would evolve higher and higher survival and fecundity would be without bound leading to unsustainable population growth. This was also stated in the final lines of the section Fitness in the SI. In the resubmission, we make this clear in the main text as well. We decided that a sensible way to introduce density dependence was to follow the advice of Mylius and Diekmann (1995), Ref. 34 in the main text of our original submission. In their Appendix, Mylius and Diekmann (1995, p. 223, first column) note that when fitness is equated with r , density dependence should be introduced via proportional changes in survival (additive changes in mortality) at all ages, while when fitness is equated with R_0 , density dependence should be introduced via proportional changes in fecundity at all ages. This is exactly what we do in our model both in the original submission and in the present re-submission.

But we fully agree with Reviewer 2 to the extent that using a form of density dependence may ring a bell in the reader, who may find useful a more open discussion of why the recent debate on extrinsic mortality cannot be of direct relevance for our work and that introducing density dependence is needed to have a realistic model.

3. COMMENT. Mathematically, the worrying aspect of the present model is that there is a deviation from the classic assumption that a completely non-senescent organism should have an exponentially distributed lifespan. Instead, the authors assume a maximum age beyond which survival is impossible. That to me is actually an a priori assumption of ageing; so I spent quite some time looking for what the authors here even consider as their operational definition of ageing, and eventually found it in the supplementary material. (A side issue: for the cognitive load placed on a reader, not having a clear definition in the main text non-ideal.) This makes me doubt the bold claims made as a whole. The issue is that they actually assume, a priori, that ageing occurs, if we define ageing in a broad sense (here, no survival chances at all beyond a specific age); the question then becomes: assuming this 'unavoidable' part of ageing is there, does this have an impact on that part of ageing that is free to evolve and that is said to occur according to definitions in the supplementary section 1.2? This way of phrasing the question reveals what might be driving much of their results: an old organism is a priori forced to have only little of its life left. Importantly, this limit is not present in the case in a memoryless process of exponentially distributed lifespans (the reason why Hamilton considers integrals and sums that go all the way to infinity).

REPLY: The fact that we imposed a maximum age was an important aspect of the analysis to which, we fully agree, the original submission was not paying enough attention. We thank the Reviewer for placing emphasis on this. In the original submission, we implicitly relied on the fact that the maximum age was arbitrary and not set to a specific number, therefore it could be made as large as desired. We thought this would essentially solve the problem of assuming a form of ageing. Following the Reviewer's comment, we have decided not to rely any longer on this heuristic solution. We changed our mathematical approach now such that no maximum age is assumed. This led to a nearly complete rewriting of the manuscript and the SI. The reassuring news is that all our key results transferred smoothly to the new setting with infinite ages implying that our results were independent of the assumed maximum age.

In the process, we had to introduce some technicalities to move from a dynamical system with an arbitrarily large, yet finite number of dimensions to an infinite dimensional dynamical system. In a few subsections of the SI and for numerical computations and figures (it is impossible to compute or to plot quantities for an infinity of ages), we have left, and explicitly acknowledged, the presence of a finite number of ages was required to derive some closed form results or some simulations that are still of theoretical interest.

We should also stress that, by introducing no finite cutoff for age, we have gone more general than the assuming an initial "non-senescent organism [with] an exponentially distributed lifespan." A strength

of our new submission is that the initial distribution of survival and fecundity over the infinity of ages can follow just any pattern, and not only constant survival and constant fecundity, as long as it satisfies Euler-Lotka equation at the demographically stationary regime.

Concerning the definition of ageing, thanks for noting that, in the original submission, we were implicitly relying too much on the first line of the introduction (“ageing is a degeneration in the physiological state of adult individuals that progressively curbs their survival and fecundity as their age increases”) of the main text to define ageing. In the resubmission, we have added to this a more formal definition to specify that ageing is defined for adult individuals as survival and fecundity decreasing with age.

4. COMMENT. So, to me, the authors seem to have shown that certain results differ from classical theory because of a combination of having chosen a life history with a finite lifespan (thus they presuppose ageing in reality, although not necessarily using their narrow definition given in the supplementary), and constrain their view to one specific form of density regulation. They don’t write about these effects at all, however, perhaps not realizing that these choices can be important; they instead write in a very feisty ‘grandspeak’-like manner accusing others of assuming what was to be proven – when they in reality haven’t done so. I believe, therefore, that they may have misidentified the reasons behind any discrepancies.

REPLY: If we understand correctly, this comment by Reviewer 2 is to the effect that our assumptions of density dependence and a finite number of ages are the true, yet covert, factors behind the discrepancies found between our model and results and those of Hamilton. And we would use these possibly artefactual discrepancies to unjustly disparage the work of Hamilton using inappropriate tones.

To address this comment, it would seem useful to first list here these discrepancies. The discrepancies between the assumptions in our model and those of the classical theory are:

- (a) the classical theory is static, as it does not contain dynamical equations, while our model is dynamic, as it contains dynamical equations;
- (b) the classical theory is by and large density independent, while our model assumes density dependence;
- (c) the classical theory assumes proportional genetic effects on survival and additive effects on fecundity, while our model assumes that any genetic effect is possible on both survival and fecundity;

The discrepancies between our results and those of the classical theory are:

- (d) the classical theory predicts that ageing is inevitable in evolution because the force of selection always declines – our model predicts that ageing is inevitable in evolution although the force of selection may not always decline.
- (e) the classical theory considers that the force of selection always declines, our model predicts that the force of selection is sure to decline at any equilibrium where ageing is present, while out of equilibrium it may or may not decline with age.

About (a)-(b): This is a modelling choice that does not depend on our original (now removed!) assumption of a finite number of ages. A static model is more limited because it is generally silent about long term outcomes of the evolutionary process. The classic theory is justified in being static for the reason that, if selection were always weakening with age, then it would represent a persistent bias against late life that would inevitably lead to the evolution of ageing regardless of initial conditions. We explain this in the new submission. But when we accept that selection may realistically not decline with age (again, not a point we made, yet our starting point), there is not any more a persistent orientation towards ageing and one needs to find out whether increasing selection can persist and perhaps lead evolution to something other than ageing. We see it as an important feature of our model that it is dynamically explicit. As explained in our reply to Comment 2 by Reviewer 2, assuming density dependence is both necessary for any long term look at evolution, as exponential population growth resulting from density independence is unsustainable, and realistic in most cases as recognized by classic authors like Hamilton, Williams and Medawar. It is, however, entirely true that we find that proportional effects on survival (additive effects on mortality), as assumed by Hamilton, it is not compatible with

an equilibrium in our model. And this is likely to depend on the specific form of density dependence we use. We specify this clearly in the resubmission. But we should recall that Hamilton does not perform dynamical and equilibrium analysis. So this is not really a discrepancy, but merely the finding of what would happen if Hamilton’s model were coupled with a certain form of density dependence.

About (c): As explained in our reply to Comment 1 by Reviewer 2, the idea of looking at the selection force by departing from the assumption of proportional effects on survival and additive effects on fecundity should be credited to Baudisch (2005) and her careful analysis of Hamilton’s 1966 work. This is our reporting of a known result and not a gratuitous accusation made by us against Hamilton, as the Reviewer suggests when saying that we are “accusing others of assuming what was to be proven – when they in reality haven’t done so.” We start our work by agreeing with Baudisch’s insightful critique of Hamilton and we work out all consequences of and probe her insight. Our original (now removed!) assumption of a finite number of ages and our use of density dependence have nothing to do with this alleged accusation. We also believe that, after Baudisch (2005), the burden of proof that Hamilton had not, unknowingly or otherwise, assumed what was to be proven, i.e. the force of selection always declines whenever one assumes proportional genetic changes in survival and additive changes in fecundity, is not on us. In our view, the Reviewer’s “complaint” about “the authors being uncharitable towards the cited work? I believe, sadly, yes,” would apply precisely if we would have failed to report the results of the work of Baudisch (2005) we cited. So we will not address this point any longer other than crediting, and redirecting the reader to, Baudisch (2005) even more forcefully in the new submission.

About (d)-(e): Here the discrepancy is on the fact that the same conclusion is reached via different routes. The fact that in the classic theory the force of selection always decline is again an assumption, as pointed out by Baudisch. Our model is more general than the classic theory because it shows that, even without that questionable assumption, ageing is the stable outcome of evolution and, at this equilibrium, one finds an always declining selection force, while not necessarily outside of it. This is fully confirmed even once we remove the assumption of a finite number of ages and let ages be infinite in number, as we do in the present re-submission. We would even dare to say that (d) is not too much of a discrepancy, after all, it is rather a vast generalization.

In the light of the fact that our analysis is essentially unaffected by introducing infinite age classes, that some form of density dependence on population growth is necessary in the long run, and that we are reporting a well grounded critique to Hamilton’s approach rather than being uncharitable towards him, we stand with our model, results and citation of Baudisch and against Comment 4 by Reviewer 2.

That said, we agree with Reviewer 2 that, stylistically, certain sections of the manuscript should be toned down compared to our initial submission to emphasize how our work expands and generalize that of Hamilton by overcoming its limitation highlighted by Baudisch. We do this in the re-submission to better situate our work within the literature. We should also add that we are truly grateful to Reviewer 2 for pushing us to look at the case of infinite ages. Figuring out the results for infinite ages give us much more confidence in the strength of our results. Thanks!

5. COMMENT. Finally, some comments about the quality of writing. I must say that despite a very extensive and thorough supplementary material, the paper as a whole is not clearly written. The schematic view provided in Figure 2 remains largely a mystery to me; I can see (and understand) 3 different regions in each oval but I don’t know what the vertical dimension is meant to represent, or what happens (= what changes are observed in the dynamics) if one moves horizontally while remaining in the no-ageing region; I also don’t know what the big blob in the top right figure signifies. I know this figure is meant to be schematic/conceptual only, but it should ideally help to understand the issues, not cause further bafflement.

REPLY: Yes, that figure was very hard to read. Thanks for noting this. We have now completely different and less esoteric figures in the resubmission.

6. COMMENT. An unfortunate feature of the MS is that the key result is derived using functions f which are themselves not explained very well; when we’re told that they are ‘generic functions that capture

any possible mutational effect on survival and fecundity at each age’, then I can see what is being said on line 112, but the following claim I do not. I reread Hamilton twice now and still I do not understand where Hamilton is assuming that f_p is the natural logarithm. Of course, one shouldn’t refrain from criticising classics, but when making bold claims it should be easy to see what one is talking about (it doesn’t help that f in Hamilton’s notation differs from the present authors’ choice).

REPLY: The way we introduced those functions was indeed quite poor and has also confused another reviewer. We now introduce them differently and, more in general, our methods are explained more at length. Concerning Hamilton, he did not use this f -notation to express different mutational effects. He assumed right away that these effects are proportional on survival and additive on fecundity. So there is no obvious way of using his notation to express our concepts for the reason that his notation would not be sufficient. In our notation, his assumption translates into $f_P(p_j) = \ln p_j$ and $f_M(m_j) = m_j$. The correct correspondence with Hamilton’s gradients can be seen by developing the right hand side of our Eqs. SI.14 and SI.20 by taking the derivatives $f'_P(p_j) = 1/p_j$ and $f'_M(m_j) = 1$ as using them as follows

$$\frac{1}{f'_P(p_j)} \frac{H_j|_{\lambda=1}}{p_j} = H_j|_{\lambda=1} = \frac{1}{T} \sum_{i=j+1} l_i m_i \lambda^{-i} \quad (3a)$$

$$\frac{1}{T} \frac{l_j}{f'_M(m_j)} = \frac{l_j}{T} \quad (3b)$$

which correspond to Eqs. 8 and 25 in Hamilton (1966), which are usually reported as his key results, e.g. Eqs. 1.6 and 1.7 in Rose (1991), with the assumption of a stationary population ($\lambda = 1$).

7. COMMENT. There are also numerous claims that I only understood once I took an “OK, if you want to express it that way, then I guess technically you can” approach - when ideally it should have been clear immediately. For example, they rather weirdly appear to complain (lines 159-160) about the classic model having an age-specific effect limited to a single fitness component, only to admit that they themselves make the same assumption. This makes me feel like countering that Hamilton did include a model where several ages (from a particular age onwards) experience elevated mortality. Thereafter, my “OK I guess this is OK after all...” moment was about realizing that “age-specific effects” can technically be interpreted broadly to include “from age i onwards” type effects. But I hope you see where I’m coming from - a more natural interpretation of a statement like that is to assume you’re talking about an effect on vital rates that occurs at one age and only that one age.

REPLY: In retrospect, this was definitely poor expression from our side. We wanted to study age-specific effects meaning exactly effects that are limited to a single age, as presupposed by the two equations above. Lines 159-60 that Reviewer 2 mentions in the original submission were: “In the classic model, as in ours, mutations have age-specific effects limited to a single fitness component, either survival or fecundity.” We did not want to express this as a complaint from our side on the assumption of age-specific effects, as Reviewer 2 proposes that this sentence could read. This sentence was merely a starting point to discuss where the observed lack of ageing in some species may come from. But in the new submission, we now specifically say that we assume age-specific effects as in the classic model. Thanks for noting this!

Reviewer 2 is totally correct in noting that Hamilton (1966) also considered lasting effects after the age of onset instead of age-specific effects. However, for some reason this other model of his has not found as much success, as, e.g., Abrams (1991, p. 334) lamented 20 years ago already, with the situation basically unchanged since, so that Hamilton’s key results are always considered to be the two equations above for age-specific effects. Therefore, we focus on this sort of effects.

In the new submission, we have adopted a clearer language throughout to explain the assumptions and the working of our model.

Reply to Reviewer 3

In this manuscript the authors propose a dynamic evolutionary model that has generic functions, f_P and f_M , capture the way mutations affect survival and fecundity. They find that in general, not just for the assumptions made in the classic approach towards the evolution of aging, declining force of selection (in the classic setting) is a result of aging rather than the necessary condition for aging to evolve.

This is a lucid, clear, and in particular well-written manuscript. One can see that the authors have not only spent a lot of thought on the issue, but have also spent great effort on presentation and visualization. The language is crystal clear. The manuscript makes the reader re-think ingrained assumptions about the evolutionary theory of aging — a much needed approach.

My comments are minor – three are relatively major but do not as such diminish the validity of this work.

We are grateful for your positive appraisal of our work!

1. COMMENT: It is in fact well known that the way the force of selection declines over ages is a function of aging (as well as the driving force of aging in the classic approach). Thus, while I appreciate the novelty of the model and the generality of f_P and f_M , the presentation could be tuned down a little bit on the decline of the force of selection being a result of aging. To be clear, I do not think that the authors wish to sweep this fact under the rug; rather, it is a matter of emphasis, of presentation. And indeed I agree that this is something that the classic theories have never really dealt with: if there is a positive feedback loop, then why does the rate of aging, however defined, not escalate without bound?

REPLY: Indeed, as another Reviewer has also noted, the overall tone of the manuscript should be lowered. We do that in the new submission. Concerning the force of selection in particular, we fully agree with Reviewer 3 that this force is known to be, in general, a function of the life history. In the new submission we clarify this point by distinguishing between the pattern that the selection force exhibits over age and the magnitude it has at each age. In the classic theory (proportional effects on survival and additive effects on fecundity), the pattern of the selection force, a declining function of age, is universal, although the exact rate of this decline may vary, for example by exhibiting a more or less pronounced decline, whence, e.g., the whole debate about the evolution of the rate of ageing in response to extrinsic mortality. After Baudisch (2005) it is instead clear that not even the pattern is universal when effects on survival are not proportional or effects on fecundity are not additive. Thanks for noting that we failed to make this distinction. In the resubmission, we make clear that our study focuses on the pattern and not on the rate of decline the force of selection exhibits in the classic theory. Our results are about the long term behaviour of the dynamical system we describe to eventually reach the same pattern of declining force of selection with age irrespective of initial conditions and form of the genetic effects. We are not concerned with the magnitude of the selection force at each age or its the rate of decline.

2. COMMENT: Similarly, it is already known that different assumptions about the way mutations affect survival and fecundity lead different conclusions (refs 10 and 27 of the manuscript, for example). Thus, what the authors add with their manuscript is the very nice dynamic picture of it all. It would be good if this were fleshed out a little more in the discussion.

REPLY: In the new submission, we have expanded the Discussion Section along these lines also in the direction described in our reply to the previous comment.

3. COMMENT: There has been some recent work by Levin and Levy, The Biostatistics of Aging: From Gompertzian Mortality to an Index of Aging-Relatedness, that also tries to overcome some of the limitations of the classic approach that the authors of the current manuscript criticize. In particular, they apply the causal pie model of causation (typing “causal pie model aging” in Google Scholar gives quite some results, in case the authors are not familiar with this concept) to investigate how the interaction of small changes affects the picture (objecting to this idea of small additive changes that underlies the classic approach). The authors might find it interesting to discuss this in regards to their work.

REPLY: Thanks for pointing us to a relevant piece of literature we had ignored. We have gone through several sections of the suggested book. While we find it of great interest per se, we were unable to

connect it directly with our work. It seems to us that the overall approach of Levin and Levy is meant to apportion certain mortality curves to its different underlying causes, only some of which may be remote/ultimate in the evolutionary sense (à la Mayr). In our model, we are unable to properly split up the different causes, evolutionary and not, leading to the final, equilibrium distribution of mortality over age. We cannot say much else other than this distribution is the result of genetic and ecological factors that are continuously mixed up at every time step along evolution, which is somewhat trivial. As we mentioned in our reply to Comment 1, our study is qualitative in a sense, as we are concerned with the overall pattern (increasing/decreasing) displayed by the trajectories of mortality, fecundity and their respective selection gradients. We do not study the exact shape of these trajectories at equilibrium. But this may be an interesting future project!

4. COMMENT: I do not think that the authors follow entirely through on their own argument. Regarding observed patterns in nature, in their (very brief) discussion, the authors state: Ageing is only one of the many patterns that these trajectories can follow. (lines 141-142). Then they remind the reader that their own results give aging as the only stable endpoint (lines 156-157). They then conclude, without much qualification, that trade-offs must be the missing part of the puzzle. I understand that this is convenient, the G matrix could do this, as the authors point out (although in their model it is a scalar multiple of the identity), but some might say that the latter is at least as much an ad-hoc assumption as Hamilton's descending selection gradients! Why trade-offs? Why not other constraints? Indeed, the authors consider only iteroparity and a constant G matrix, while the discussion of the manuscript seems to take a more general view. And evolution is never finished, so would the authors not be as bold as to speculate that the non-aging species currently observed are transient, at least for iteroparous species?

REPLY: In our Discussion, the fact that we appear to jump to the trade-off hypothesis, once age-specificity is deemed insufficient to explain observed variation, is due to our reliance on the overview of the problems with Hamilton's approach offered by Caswell and Salguero-Gómez (2013, p. 586), which is ref. 13 in the original submission. According to them,

“Hamilton's theory has been the subject of intense discussion, and two ways to avoid the conclusion of inevitable senescence have been noted. One is to assume some other kinds of trade-offs; for example, between mortality and fertility, or generated by allocation of energy (Tuljapurkar 1997; Baudisch 2008). Another is to focus on traits that modify mortality or fertility in other ways (Baudisch 2005); the selection gradients on traits that produce proportional changes in mortality or fertility are not necessarily monotonically decreasing with age.”

We do not really assume that trade-offs will do the job of explaining absence of ageing. On the contrary, this is a conclusion of a reasoning. Starting from the above quotation, we have investigated to the best of our abilities the second way mentioned therein: that survival and fecundity may change in ways not contemplated by Hamilton. According to our results, this way is not conducive to anything other than ageing anyway. Hence, it would seem that only the first way (trade-offs) would be viable. In the resubmission, we try to make this reasoning more explicit.

The distinction between trade-off and constraint that the Reviewer mentions is subtle and potentially slippery, e.g. Roff and Fairbairn (2007), to the point that authors argue that it is generally safe to only speak of trade-off without differentiating them from constraints (Garland, 2014). This is what we do here as we do not feel that our manuscript is the right place to open up a discussion on this distinction.

It is true that we are only concerned with iteroparity. Since it would seem that semelparity is a form of extremely fast ageing Kirkwood and Austad (2000), we would argue that the case is implicitly covered as well by our finding that ageing is a stable evolutionary outcome. As stated in our reply to the previous comment, we are concerned with the pattern of ageing and not its magnitude or rate.

Concerning the possibility that the observed absence of ageing may be a transient phenomenon, we agree that this could be the case. We specify this in the new submission. However, we also specify that we are not in the position, i.e. our model alone does not entitle us, to tell whether the observed absence of ageing is transient or whether trade-offs stabilize it for the very reason that we have not explored in

our work the role of trade-offs. We can only get to a dilemma, which we now make in the resubmission: either the observed lack of ageing is transient, or it is stabilized by trade-offs. But we also add that we tend to favour the second horn of this dilemma because there are theoretical results, those that we mention in our refs, about ESS life history strategies with absence of ageing. More in general, it would seem to be more likely to observe stable phenomena rather than unstable ones precisely because the former are lasting while the latter are not if they arise at the same rate.

5. COMMENT. Lines 144-146: “This is questionable, as the same theoretical argument that is made to derive this principle leads to a contradiction of it under slightly different assumptions.” Not sure what is meant here. What is the contradiction? Reference 10 given, Baudisch PNAS 2005, calculates selection gradients for non-additive perturbations. Reference 27, also from Baudisch’ group, shows that the same can be achieved by making the perturbations (rather than the selection gradients) proportional to mortality, after which the product of the two is integrated out to give the fitness effect. So it seems to be more about the biological argument rather than the choice of mathematical representation of the biologics: biologics, not mathematics, make aging inevitable or not. Might this be what the authors mean? (MJ Wensink, TF Wrycza, A Baudisch - PloS one, 2014 seems to make an argument somewhat akin)

REPLY: That was bad phrasing from our side. The contradiction we mentioned was the fact that the stated inevitability of the decline in the selection force turns out not to be such upon minimal changes in the assumptions, as the ref mentioned by Reviewer 3 shows. This is not a genuine logical contradiction (reaching a false conclusion using valid arguments from a given set of premises), as it involves changing one premise (form of mutational effect). In the new submission, we avoid this formulation.

6. COMMENT. Caswell proved that Hamilton’s selection gradients can be written as the product of reproductive value and the stable age distribution. Since the authors introduce these concepts in the appendix, it might be nice to remark on this there. I think the reference is <https://www.jstor.org/stable/26349604> (but please check).

REPLY: Thanks for the suggestion. However, we must say that while Caswell (1978) did prove this in terms of general matrix models representing any demography and not only the age-based case, (the paper mentioned by Reviewer 3 presents the continuous time version for the age classified case) the first derivation of Hamilton’s selection gradients in terms of age class distribution and reproductive value seems due to Goodman (1971), see in particular his Eqs. 34, 36 and 39. But given that we do not explicitly introduce the expression for the stable age distribution, we would prefer to avoid adding more formulas to our already lengthy SI.

7. COMMENT. Probably not and I haven’t thought this through, but could the generality of f_P and f_M confound with the variance of the G matrix in predicting the evolutionary dynamics, at least in the short term, if G evolves? There seem to be unlimited degrees of freedom for both, so I suspect there might be some cancelling out to do, at least by approximation, taking first Taylor terms, something like that.

REPLY: If we understand correctly this comment, Reviewer 3 is pointing to the possibility that it remains an open question what it would happen, evolutionary, if we would also let the genetic covariance matrix G evolve rather than taking it to be a constant scaled identity matrix. We suspect that this would lead to complicated interactions between the selection gradient and the functions. As we briefly mention in the discussion section of the original submission as well as in the new submission, the evolution of the G matrix is a very complicated topic that is not easily to handle analytically. The lengthy SI of our work testifies that also working with the f_P and f_M functions alone is not too straightforward. All in all, we are afraid we cannot at this stage get a clear picture of how to merge our approach with an evolving G matrix and, even less, can we envisage what the resulting outcome would be. But we now state this openly in the Discussion.

8. COMMENT. (The appendix is a small book I hope that I am forgiven for not checking the equations line by line, but it certainly looks tidy and considered!)

REPLY: Thanks. We should however inform you that we heavily revised the SI in the new submission in favour of a much more general version of our model where we remove the assumption of our initial

submission of an arbitrarily large, yet finite maximum age so that there is no maximum age and potentially there are infinite ages.

References

- Peter A. Abrams. The fitness costs of senescence: The evolutionary importance of events in early adult life. *Evolutionary Ecology*, 5(4):343–360, Oct 1991. ISSN 1573-8477. doi: 10.1007/BF02214152. URL <https://doi.org/10.1007/BF02214152>.
- Peter A Abrams. Does increased mortality favor the evolution of more rapid senescence? *Evolution*, 47: 877–887, 1993.
- Annette Baudisch. Hamilton’s indicators of the force of selection. *Proceedings of the National Academy of Sciences USA*, 102:8263–8268, 2005.
- Hal Caswell. A general formula for the sensitivity of population growth rate to changes in life history parameters. *Theoretical Population Biology*, 14(2):215 – 230, 1978. ISSN 0040-5809. doi: [http://dx.doi.org/10.1016/0040-5809\(78\)90025-4](http://dx.doi.org/10.1016/0040-5809(78)90025-4). URL <http://www.sciencedirect.com/science/article/pii/0040580978900254>.
- Hal Caswell and Roberto Salguero-Gómez. Age, stage and senescence in plants. *Journal of Ecology*, 101 (3):585–595, 2013. ISSN 1365-2745. doi: 10.1111/1365-2745.12088. URL <http://dx.doi.org/10.1111/1365-2745.12088>.
- Thomas Flatt and Linda Partridge. Horizons in the evolution of aging. *BMC Biology*, 16:93, 2018.
- Theodore Garland. Trade-offs. *Current Biology*, 24(3):R60–R61, 2014.
- Leo A. Goodman. On the sensitivity of the intrinsic growth rate to changes in the age-specific birth and death rates. *Theoretical Population Biology*, 2(3):339 – 354, 1971. ISSN 0040-5809. doi: [https://doi.org/10.1016/0040-5809\(71\)90025-6](https://doi.org/10.1016/0040-5809(71)90025-6).
- W.D. Hamilton. *Narrow Roads of Gene Land*, volume 1: Evolution of Social Behaviour. Freeman Spektrum, New York, 1996.
- Thomas B. L. Kirkwood and Steven N. Austad. Why do we age? *Nature*, 408:233—238, 2000.
- Peter Brian Medawar. *The uniqueness of the individual*. Basic Books, New York, 1958.
- Sido D. Mylius and Odo Diekmann. On evolutionarily stable life histories, optimization and the need to be specific about density dependence. *Oikos*, 74(2):218–224, 1995. ISSN 00301299, 16000706. URL <http://www.jstor.org/stable/3545651>.
- Derek A. Roff and D. J. Fairbairn. The evolution of tradeoffs: where are we? *Journal of Evolutionary Biology*, 20(2):433–447, 2007. doi: 10.1111/j.1420-9101.2006.01255.x. URL <https://onlinelibrary.wiley.com/doi/abs/10.1111/j.1420-9101.2006.01255.x>.
- Michael R. Rose. *Evolutionary biology of aging*. Oxford University Press, New York, 1991.
- George C. Williams. Pleiotropy, natural selection, and the evolution of senescence. *Evolution*, 11(4):398–411, 1957. ISSN 00143820. URL <http://www.jstor.org/stable/2406060>. 10.2307/2406060.

Reviewers' Comments:

Reviewer #1:

Remarks to the Author:

I will not review the paper in detail here. My concerns from the first version have been adequately addressed, and I continue to believe that this paper makes a valuable contribution to thinking about the evolution of ageing, and should be published. There have been significant changes to some of the mathematical analysis. I have tried to examine with some care the material that seemed new, but have not rechecked results that seem essentially unchanged from the first version.

There are sporadic problems with grammar and syntax (e.g., line 172 of main text), so the whole would benefit from another careful round of proofreading.

Reviewer #2:

Remarks to the Author:

Apologies that I had to ask for extra time to re-review this paper, the revision came at an unusually bad time for me. Now, I am very happy with the extensive revisions that the authors have done here, and the effort they put into explaining their views in the response letter. I also am happy to accept that I was wrong with my hunch that it was the assumption of finite maximum age that was driving the results!

I only have very few remaining comments. First, the paragraph on lines 43-56. This is a really important paragraph to make the reader understand the central premise of the study, so it would be good to make it crystal clear what is meant here (and if it requires making it 2 paragraphs not 1, perhaps that could be done). Words such as 'ecology' can mean a lot of different things in different people's minds, and 'dynamics' is similarly a word that has a specific meaning here: I think the authors use it synonymously with "when the life history changes, we need to re-evaluate the forces of selection as well". Since dynamics broadly speaking only means that something is changing over time, it appears worthwhile to specify the precise meaning that the word is used in the current context. As a whole, unpacking the statements made here could improve readability a lot.

Also, even though the meaning of m_{j+1}/m_j is unambiguously defined, I think something could be done to fig.2 to avoid the wrong first impression that the dots refer to fecundity itself which bounces up and down in fig. 2a. Empirical findings on senescence almost always plot that, not the ratios depicted here. It might even be worth depicting the same outcome twice: once like it is done here, and below or above it, the actual fecundities associated with the red dots so that one can see that they decline with age.

Once again, sorry I was late with this review.

Reply to Reviewers

We are grateful to the Reviewers for having found merit in our revised submission and for pointing out issues that still need to be addressed. We now submit a new version that addresses all points that were made.

Reply to Reviewer 1

I will not review the paper in detail here. My concerns from the first version have been adequately addressed, and I continue to believe that this paper makes a valuable contribution to thinking about the evolution of ageing, and should be published. There have been significant changes to some of the mathematical analysis. I have tried to examine with some care the material that seemed new, but have not rechecked results that seem essentially unchanged from the first version.

We are truly glad that your positive assessment of our work has remained unchanged!

1. COMMENT: There are sporadic problems with grammar and syntax (e.g., line 172 of main text), so the whole would benefit from another careful round of proofreading..

REPLY: We have run a careful proofreading to minimize such problems. In particular, we have broken overlong sentences (like line 172 of of main text in the previous submission) into smaller and simpler blocks.

Reply to Reviewer 2

Apologies that I had to ask for extra time to re-review this paper, the revision came at an unusually bad time for me. Now, I am very happy with the extensive revisions that the authors have done here, and the effort they put into explaining their views in the response letter. I also am happy to accept that I was wrong with my hunch that it was the assumption of finite maximum age that was driving the results! I only have very few remaining comments.

We are grateful that you raised those comments in the previous round of review, as they were of guidance to improve our submission.

1. COMMENT: First, the paragraph on lines 43-56. This is a really important paragraph to make the reader understand the central premise of the study, so it would be good to make it crystal clear what is meant here (and if it requires making it 2 paragraphs not 1, perhaps that could be done). Words such as ecology can mean a lot of different things in different peoples minds, and dynamics is similarly a word that has a specific meaning here: I think the authors use it synonymously with when the life history changes, we need to re-evaluate the forces of selection as well. Since dynamics broadly speaking only means that something is changing over time, it appears worthwhile to specify the precise meaning that the word is used in the current context. As a whole, unpacking the statements made here could improve readability a lot.

REPLY: Yes, that paragraph was leaving too much implicit. We have now rewritten and expanded upon this paragraph to make things clearer. In particular, we do not rely anymore on some implicit notion of dynamics that the reader should already have. Instead, we say explicitly that we track changes in age-specific fecundity, survival and the selective forces acting upon them. We also say explicitly that this is what one needs to do when the assumptions of Hamilton's classic theory are relaxed in order to understand whether ageing evolves or not.

2. COMMENT: Also, even though the meaning of m_{j+1}/m_j is unambiguously defined, I think something could be done to fig.2 to avoid the wrong first impression that the dots refer to fecundity itself which bounces up and down in fig. 2a. Empirical findings on senescence almost always plot that, not the ratios depicted here. It might even be worth depicting the same outcome twice: once like it is done here, and below or above it, the actual fecundities associated with the red dots so that one can see that they decline with age.

REPLY: We agree that the y -axis in figs. 2 and 3 of our last submission with the ratio of successive age-specific quantities could be unusual for the readership. Following your suggestions, we have tried in different ways to change those figures by adding panels with the usual axes to report equilibria. However, we were unsatisfied with the outcome every time. The problem is that there is no obvious way of visualizing the existence, number and stability, or our ignorance thereof, for these equilibria. For this reason, we have opted for modifications of the original figures by making more explicit where the "ageing region" is and how fecundity and survival at two successive ages compare when we consider the region above 1 and below 1. We admit this may be not the optimal solution, but we were unable to come up with a better one.